# Comparing lagged impacts of mobility changes and environmental factors on COVID-19 waves in rural and urban India: A Bayesian spatiotemporal modelling study

Eimear Cleary[1,*], Fatumah Atuhaire[1], Alessandro Sorichetta[2], Nick Ruktanonchai[3], Cori Ruktanonchai[3], Alexander Cunningham[1], Massimiliano Pasqui[4], Marcello Schiavina[5], Michele Melchiorri[5], Maksym Bondarenko[1], Harry E R Shepherd[1], Sabu S Padmadas[6,7], Amy Wesolowski[8], Derek A T Cummings[8,9], Andrew J Tatem[1], Shengjie Lai[1,10]

1 WorldPop, School of Geography and Environmental Science, University of Southampton, Southampton, United Kingdom, 2 Department of Earth Sciences "Ardito Desio", Università degli Studi di Milano, Milan, Italy, 3 Department of Population Health Sciences, Virginia-Maryland College of Veterinary Medicine, Virginia Tech, Blacksburg, Virginia, United States of America, 4 Institute for Bioeconomy, National Research Council of Italy (IBE-CNR), Rome, Italy, 5 European Commission, Joint Research Centre, Ispra, Virginia, Italy, 6 Department of Social Statistics & Demography, Faculty of Social Sciences, University of Southampton, Southampton, United Kingdom, 7 Department of Public Health & Mortality Studies, International Institute for Population Sciences, Mumbai, India, 8 Department of Epidemiology, Johns Hopkins Bloomberg School of Public Health, Baltimore, Maryland, United States of America, 9 Department of Biology and Emerging Pathogens Institute, University of Florida, Gainesville, Florida, United States of America, 10 Institute for Life Sciences, University of Southampton, Southampton, United Kingdom

☯ These authors contributed equally to this work.
* e.cleary@soton.ac.uk

## Abstract

Previous research in India has identified urbanisation, human mobility and population demographics as key variables associated with higher district level COVID-19 incidence. However, the spatiotemporal dynamics of mobility patterns in rural and urban areas in India, in conjunction with other drivers of COVID-19 transmission, have not been fully investigated. We explored travel networks within India during two pandemic waves using aggregated and anonymized weekly human movement datasets obtained from Google, and quantified changes in mobility before and during the pandemic compared with the mean baseline mobility for the 8-week time period at the beginning of 2020. We fit Bayesian spatiotemporal hierarchical models coupled with distributed lag non-linear models (DLNM) within the integrated nested Laplace approximation (INLA) package in R to examine the lag-response associations of drivers of COVID-19 transmission in urban, suburban and rural districts in India during two pandemic waves in 2020-2021. Model results demonstrate that recovery of mobility to 99% that of pre-pandemic levels was associated with an increase in relative risk of COVID-19 transmission during the Delta wave of transmission. This increased mobility, coupled with reduced stringency in public intervention policy and the emergence of the Delta variant, were the main contributors to the high COVID-19 transmission peak in India in April 2021. During both pandemic waves in India, reduction in human mobility, higher stringency of interventions, and climate factors (temperature and

**Data availability statement:** The code used for the analysis in this study is available at the following GitHub repository: https://github.com/e3cleary/COVID-19-INDIA.git. COVID-19 data at district level (admin II) for 666 districts from 26 April 2020 to 31 October 2021 were obtained from www.covid19india.org, a volunteer-driven, crowdsourced tracker for COVID-19 cases in India. The Google COVID-19 Aggregated Mobility Research Dataset used in this study is available with permission from Google LLC. All data were provided and analyzed in an anonymous format, without access to personally identifiable information.

**Funding:** This study was supported by the National Institutes of Health (R01AI160780) and the Bill & Melinda Gates Foundation (INV-024911 to EC). The funders of the study had no role in study design, data collection, data analysis, data interpretation, or writing of the report. The corresponding authors had full access to all the data in the study and had final responsibility for the decision to submit for publication. The views expressed in this article are those of the authors and do not represent any official policy.

**Competing interests:** The authors have declared that no competing interests exist.

precipitation) had 2-week lag-response impacts on the $R_t$ of COVID-19 transmission, with variations in drivers of COVID-19 transmission observed across urban, rural and suburban areas. With the increased likelihood of emergent novel infections and disease outbreaks under a changing global climate, providing a framework for understanding the lagged impact of spatiotemporal drivers of infection transmission will be crucial for informing interventions.

## Introduction

The COVID-19 pandemic highlighted the intrinsic role of human movement, along with demographics and environmental factors, in the dispersal of human pathogens in a highly connected, mobile and globalised society [1–3]. As the global climate changes, and environmental and extreme weather events increase in frequency, the emergence of novel zoonotic diseases and outbreaks of bacterial, parasitic and viral infections is likely to become more frequent [4]. Effective and efficient responses to future outbreaks and epidemics require a thorough understanding of the infection transmission drivers that contributed to different COVID-19 pandemic waves, and interventions that were successful in reducing transmission.

In India, the initial wave of COVID-19 was contained by a nationwide lockdown, which extended from March 31st to May 31st, 2020 [5], with a subsequent phased lockdown for containment zones in effect until June 30th, 2020 [6]. The first wave of COVID-19 transmission in India was characterised by mild clinical infection and a relatively low mortality rate of less than 3% [5]. Several serosurveys carried out following the initial pandemic wave in India determined a high proportion of asymptomatic infections [7–10], leading to speculation as to the reasons for lower incidence of severe clinical cases including population demographics and innate population immunity [11,12].

In March 2021, India experienced a severe second wave of COVID-19 transmission with a high proportion of infection associated mortality [13]. The Delta variant, or B.1.617 lineage, dominant during the second transmission wave was first identified in Maharashtra in late 2020 [14] before quickly spreading throughout India and to at least 90 other countries [15]. Compared with the initial pandemic wave in India, the Delta wave was characterised by high morbidity and mortality, even among a younger age cohort, overwhelming health systems across the country [16,17]. On April 26th 2021, India recorded 360,960 new cases, at the time the highest number of daily new SARS-CoV-2 infections recorded worldwide [18], and by mid-June 2021 more than 29 million cases of COVID-19 had been confirmed [19]. During the second pandemic wave, the number of COVID-related deaths in India ranked third globally with an estimated 2.7 million COVID-19-related deaths occurring between April and July 2021 [20].

Although reasons for the second wave of transmission were unclear, it was speculated that the surge in case numbers was attributed to the circulation of the B.1.617 lineage of SARS-CoV-2 (Delta variant), which had a more effective transmission capability, shorter incubation period and was more pathogenic than previous lineages [17,21,22]. Prior to this surge in transmission, adherence to COVID-19 preventative behaviours in India was less stringent, possibly due to pandemic fatigue, economic necessity and complacency due to the perception that clinical case infections in India were mild relative to other populations [16,23]. Population mobility, which had begun to increase relative to mobility during national lockdown interventions, including rural-urban-rural migration to mass election rallies and social and religious gatherings such as Kumbh Mela (approximately 7 million people), was also likely to be a primary driver of the second wave of SARS-CoV-2 in India [13,15].

Previous research has explored the relationship between human mobility in response to government interventions and COVID-19 transmission during the early stages of the pandemic [24–26], or state level associations between human mobility and COVID-19 transmission during the Delta pandemic wave [27]. However, to our knowledge, no previous research has explored the impact of inter-district movement across both pandemic waves, relative to pre-pandemic mobility levels, using fine spatial scale aggregated mobility data on COVID-19 transmission in India. The contribution of district level urbanisation [28,29], population density and demographics [30,31], climate [28,32] and stringency of government interventions [24] to COVID-19 transmission in India has also previously been investigated. However, methodological approaches have included simple correlation [24,30] or regression analyses [33] and, to the best of our knowledge, no spatiotemporal modelling approach has been used to explore the urban-rural district level associations of human mobility, stringency of government interventions, and climate with transmission risk across both pandemic waves in India.

Further to this, although extensive research has been conducted exploring the impact of various climate drivers on COVID-19 transmission, substantial heterogeneity exists in published results. For example, non-linear associations have been found between temperature and global COVID-19 transmission [34], with lower temperature negatively associated with daily COVID-19 cases in a study among 127 countries [35]. Elsewhere, across 154 different countries, higher temperatures have been found to be negatively correlated with COVID-19 [36]. Globally, the ultraviolet (UV) index has been negatively associated with COVID-19 transmission [37,38], with a one to two week lagged impact [39,40]. In terms of the impact of humidity and precipitation on COVID-19 transmission, results of previous published literature are varied and inconsistent, with some research indicating a negative impact of humidity [41,42] and a positive relationship between precipitation and COVID-19 cases [41]. Elsewhere, a weak association between cumulative precipitation [43] and no significant correlation between humidity and COVID-19 [44,45] has been observed.

While COVID-19 is no longer classed as a public health emergency, it remains a pandemic with significant associated mortality, long term health effects and seasonal transmission peaks [46–48]. In order to prepare for seasonal COVID-19 epidemics, and plan allocation of resources such as testing and vaccination booster campaigns, it is critical to develop a framework for exploring spatiotemporal variations in drivers of transmission across urban, suburban, and rural areas. In this study, we quantified changes in mobility patterns and travel networks across India, before and during the COVID-19 pandemic, using spatially resolved, aggregated and anonymized weekly human movement datasets obtained from Google. We used a Bayesian spatiotemporal hierarchical framework, coupled with distributed lag non-linear models (DLNM) to examine the lag-response associations between the transmission dynamics of COVID-19 and drivers of transmission during the initial wave (July to November 2020) and Delta wave (March to July 2021) of SARS-CoV-2 in India. We also compared the lagged impacts of mobility metrics, climate covariates, and stringency of government interventions on the transmission of SARS-CoV-2 lineages between both pandemic waves, and across urban, suburban and rural delineated districts.

## Methods

### Ethics statement

Ethical clearance for collecting and using secondary data in this study was granted by the institutional review board of the University of Southampton (No. 61865). All data were

supplied and analysed in an anonymous format, without access to personal identifying information.

## Data sources

**COVID-19 incidence data.** In India, administrative units are divided into state (36 including eight union territories), district and township, corresponding to spatial administrative levels I, II and III, respectively (S1 Fig). The daily number of confirmed COVID-19 cases at country level were obtained from the COVID-19 Data Repository assembled by the Centre for Systems Science and Engineering (CSSE) at Johns Hopkins University [49]. We also obtained COVID-19 data at district level (admin II) for the period of 26 April 2020 to 31 October 2021 for 666 districts from www.covid19india.org, a volunteer driven, crowdsourced tracker for COVID-19 cases in India [50]. COVID-19 data were available in 666 district units, as in some cases, depending on testing capacity and guidelines in each federal state, data were aggregated to state level only or case incidence was estimated by state pool [50].

Administrative level I and II shapefiles for India, corresponding with state and district level, were obtained from the Database of Global Administrative Areas (GADM version 3.6) (https://gadm.org/). Since the last national census of population in India in 2011, new districts have been created by splitting and rearranging some administrative boundaries [30]. COVID-19 data aggregated to current district boundaries were merged with 2011 administrative level II units according to the best spatial alignment of current and previous district boundaries. For the purpose of spatial modelling, the islands in Lakshadweep and the Andaman Islands have been unified as discrete spatial areas and treated as distinct districts. The authors remain neutral with regard to jurisdictional claims in maps used in this study.

**Google COVID-19 Aggregated Mobility Research Dataset.** Aggregated and anonymized weekly human movement datasets were obtained from Google to measure changes inmobility across and within regions in India from November 10, 2019, to December 31, 2021, and to assess their impacts on COVID-19 transmission. The Google mobility dataset contains anonymized mobility flows aggregated over users who have turned on the Location History setting, which is off by default. This is similar to the data used to show how busy certain types of places are in Google Maps — helping to identify when a local business tends to be the most crowded. The dataset aggregates flows of people between S2 cells, which here is further aggregated by district of origin and destination. Each S2 cell represents a quadrilateral on the surface of the planet and allows for efficient indexing of geographical data.

To produce this dataset, machine learning was applied to log data to automatically segment data into semantic trips [51,52]. To provide strong privacy guarantees, all trips were anonymized and aggregated using a differentially private mechanism to aggregate flows over time (see https://policies.google.com/technologies/anonymization). This research is done on the resulting heavily aggregated and differentially private data. No individual user data was ever manually inspected, only heavily aggregated flows of large populations were handled. All anonymized trips are processed at aggregate level to extract their origin, destination, location and time. For example, if users travelled from location a to location b within time interval t, the corresponding cell (a, b, t) in the tensor would be $n \mp err$, where err is Laplacian noise. The automated Laplace mechanism adds random noise drawn from a zero mean Laplace distribution and yields $(\varepsilon, \delta)$-differential privacy guarantee of $\varepsilon = 0.66$ and $\delta = 2.1 \times 10^{-29}$. The parameter $\varepsilon$ controls the noise intensity in terms of its variance, while $\delta$ represents the deviation from pure $\varepsilon$-privacy. The closer they are to zero, the stronger the privacy guarantees. Each user contributes at most one increment to each partition. If they go from a location a to another location b multiple times in the same week, they only contribute once to the aggregation count.

The summed weekly domestic mobility inflows and outflows of each district were then divided by the number of origin S2 cells (each was calculated only once) that contained data between November 10, 2019 and December 31, 2021. Any potential bias that might be introduced by discarding the increasing number of S2 cells in order to protect privacy due to the decreasing number of travellers under travel restrictions was accounted for. For comparability of changes in mobility across districts, aggregated flows were further standardised using pre-pandemic mean baseline levels of mobility for the first eight weeks of 2020 (December 29, 2019 – February 22, 2020) (**S2** – **S4 Fig** & **S29 Fig**). This dataset was analysed by researchers at the University of Southampton, UK as per the terms of the data sharing agreement. Production of this anonymized and aggregated dataset has been detailed in previous studies [3,51–53].

**Stringency of COVID-19 intervention.** Stringency Index of COVID-19 intervention policy in India data were obtained from the Oxford COVID-19 Government Response Tracker (OxCGRT) project at state level and daily temporal resolution (**S5 Fig** & **S30 Fig**). The Stringency Index is a composite index of government responses to the COVID-19 pandemic compiled by OxCGRT based on data collected from publicly available sources such as news articles, and government press releases and briefings from 1 January 2020 [54,55]. The project tracks national government policies and interventions across a standardized series of indicators and creates a suite of composite indices to measure the extent of these responses to understand how government responses evolved over the course of the pandemic [55]. The Stringency Index was calculated as a composite score of 18 indicators of closure and containment, health, and economic policy [24,54]. Scores were created using an additive unweighted approach, taking the ordinal value and adding a weighted constant if the policy was general rather than targeted. The maximum values were rescaled to create a score ranging from 0 to 100, with higher scores indicating stricter measures [54]. Stringency Index data for India were obtained from 27th April 2020 to 25th July 2021.

**Climate data.** Three-dimensional Network Common Data Form (NetCDF) climate data were obtained from the Copernicus Climate Data online repository (Copernicus Climate Change Service, Climate Data Store, (2023): ERA5 hourly data on single levels from 1940 to present. Copernicus Climate Change Service (C3S) Climate Data Store (CDS), https://doi.org/10.24381/cds.adbb2d47 (Accessed on 19-05-2023). Data were ERA5 daily reanalysis global climate data obtained for January 2019 to March 2021, gridded to 0.25 degrees of latitude and longitude. Variables obtained were mean temperature of air (°C at 2m above the surface of land, sea or inland waters), accumulated precipitation (metres), relative humidity (%) and downward ultraviolet (UV, $KJ/m^2$ *per hour* ) radiation at the Earth's surface (**S6**-**S9 Fig** & **S31** – **S34 Fig**).

ERA5 data are the fifth generation of European Centre for Medium-Range Weather Forecasts (ECMWF) reanalysis for the global climate and weather for the past 4 to 7 decades. Reanalysis is a method of combining model data with global observations for producing complete and consistent datasets for a large number of atmospheric, ocean-wave and land-surface quantities. Reanalysis works in the same way as the principle of data assimilation which combines previous forecasts with newly available observations on a 12-hour basis to produce new best estimates of atmospheric measures [56]. Climate data were extracted from NetCDF files using the ncdf4 [54,57] and RNetCDF [58] packages in R statistical software version 4.1.0 and aggregated to district level using Quantum Geographic Information Systems (QGIS) software [59].

**Urban and rural classification.** Data on the degree of urban, rural and suburban spatial area within each district (admin level II) were derived from the Global Human Settlement Layer (GHSL) [60] using the Degree of Urbanisation – Territorial units classifier

(GHS-DU-TUC) tool. The GHS-DU-TUC tool classifies local units from a settlement classification grid according to the Degree of Urbanisation (DEGURBA). It operationalises the method recommended by the 51st Session of the United Nations Statistical Commission to delineate cities, urban and rural areas (stage 2, units classification) as defined by the Degree of Urbanisation levels 1 and 2. Categorised variables for each degree of urbanisation (DEGURBA_L1_1 to DEGURBA_L1_3) were generated for degree of urban vs. rural spatial area in each district area in accordance with methods for implementation of INLA models outlined in Lezama-Ochoa *et al.* 2020 [61].

Degree of urbanisation was categorised as follows: (1) Rural (mostly thinly populated areas), (2) Suburban (mostly intermediate density areas), and (3) Urban (mostly densely populated areas). Population data for 2020 were obtained at 100m spatial resolution from the WorldPop online repository (https://www.worldpop.org/) and aggregated to calculate population density per km2 for each district. Data on public holiday time periods were obtained from the National Portal of India online repository (https://www.india.gov.in/). Public holidays, which included the date of public holiday and one day before and after, were assigned a value of 1. All other days were given a value 0.

## Data analysis

*Exploring changes in mobility in India during the pandemic.* To gain a better understanding of travel networks and connectivity across India, we explored the overall patterns in domestic travel by rural, semi-urban and urban delineated areas in India, using weekly Google mobility data from November 10, 2019, to December 31, 2021. The relative levels of mobility across regions (regions are defined as six zones comprising different states in India defined under the States Reorganisation Act 1956 [62]) and weeks were further calculated for each type of flow, relative to the mean level of pre-pandemic baseline in each region from December 29, 2019, to February 22, 2020. We also defined mobility reductions and communities of population movements between administrative level II units, i.e., districts, across the country for five periods (S2 Fig & S3 Fig): 1)

Pre-pandemic period (15 weeks) from November 10, 2019 to February 22, 2020; 2) First lockdown (6 weeks), from March 22 to May 2, 2020, that included strict travel restrictions, stay-at home orders and closure of many businesses; 3) Pre-second lockdown period (8 weeks) from January 31 to March 27, 2021; 4) Second lockdown (6 weeks) for the Delta wave, from April 18 to May 29, 2021; 5) post-second lockdown period (8 weeks), from November 7 to December 31, 2021, after travel restrictions for COVID-19 had been lifted in India. In the context of travel networks, a community refers to a group of areas that are more closely connected internally than with other areas in the network [63,64]. Community structures were detected using the Louvain algorithm, a method of extracting communities from large networks [63]. We mapped the communities identified to highlight distinct geographic groupings of districts in terms of movements across periods.

**Reproduction number.** To account for variations in the transmissibility of COVID-19, we estimated the instantaneous reproduction number $R_t$, a measure of initial transmissibility of each variant, across the waves, for each district of the country with available case data (S10 Fig & S35 Fig). First, the number of daily new COVID-19 cases at district level were smoothed using a Gaussian smoothing approach over a 7-day rolling window [65]. Second, the mean incidence of cases at day *t* was assumed following the Poisson distribution that is defined as:

$$E\left(I_t\right) = R_t \sum_{k=1}^{t} I_{t-k} w_k$$

where $I_{t-k}$ is the incidence at time $t-k$, $w_k$ is the infectivity profile which depends on the serial interval of COVID-19 (5.2, 95%CI: 4.9–5.5) [66]. The serial interval represents the time between onset of the primary case to onset of the secondary case. Last, we estimated the daily $R_t$ for each district with a 7-day sliding window, using the EpiEstim package [67] in R statistical software version 4.1.0 [68].

In order to account for changing transmissibility of COVID-19 caused by different variants in the modelling, we also estimated the variant-specific basic reproduction number (R0) across the waves. The variant-specific initial reproduction number (R_in) is a measure of initial transmissibility which accounts for the effects of interventions and no depletion in susceptibility in the population. We first assembled data of the biweekly proportion of sequences of six main SARS-CoV-2 variants, including lineages B.1.1.7 (VOC Alpha), B.1.351 (Beta), P.1 (Gamma), B.1.617.2 (Delta), B.1.525 (Eta), and B.1.617.1 (Kappa), based on SARS-CoV-2 sequence data in the Global Initiative on Sharing All Influenza Data (GISAID) [69], as of 25 October 2021. Using an approach described by Ge *et al.* [70], we then calculated a weighted average of basic reproduction numbers of the six variants mentioned above and the SARS-CoV-2 strain in circulation before VOCs became predominant (seven coronavirus variants in total).

**Models for examining lag-response associations between COVID-19 transmission and different factors.** We built spatiotemporal Bayesian hierarchical models which consisted of weekly changes in the $R_t$ of COVID-19 transmission for 666 districts in India where data were available during 17 weeks from March 7 to July 3, 2021 (Delta wave) and during the 19 weeks between July 19th 2020 and November 29th 2020 (wave 1). We used a Bayesian spatiotemporal hierarchical framework to explore the drivers of infection, accounting for fixed and random spatial and temporal effects using the integrated nested Laplace approximation (INLA) approach. Bayesian spatiotemporal models provide a robust, flexible approach for exploring drivers of infection transmission, while incorporating spatial and temporal dependencies and quantifying uncertainty in predictions [71,72]. Bayesian models also allow fitting of prior parameters to incorporate prior knowledge and uncertainty in the model. The INLA approach is an alternative to Markov Chain Monte Carlo (MCMC) methods which approximates posterior estimations by applying numerical integrations for fixed effects and Laplace integral approximation to model random effects [72,73]. Bayesian spatiotemporal models were built using the INLA package in R version 4.1.0 [73].

We assumed that $R_t$ adjusted by $R_0$, denoted as $\triangle R_t = R_t / R_0$, conformed to the Gamma distribution, $\triangle R_t \mid \mu\_t \sim Gamma\left(\dfrac{\mu_t}{0.5}, 0.5\right)$, where $\mu\_t$ was the corresponding distribution expectation (or mean), reflecting the shape-rate parameterisation of the Gamma distribution used by the INLA package. A gamma distribution was determined based on the distribution of the observed data and based on the lowest deviance information criterion (DIC) during initial exploratory analyses using Weibull, Gaussian and gamma distributions in our base model [74]. Models were structured to account for spatial and temporal dependencies in the data while incorporating covariates associated with change in $R_t$.

Spatiotemporal models were constructed by defining a likelihood for the observed data (gamma distribution) and specifying latent processes to capture spatial and temporal effects. These latent processes were implemented as random effects in the model formulation. Variable selection was based on previous published literature (described above), and inclusion of covariates in Bayesian spatiotemporal models was determined by initial exploratory analyses using generalised linear models (GLMs). Drivers were included in the model as fixed effects, directly incorporated into the linear predictor alongside the random effects. INLA was employed as the computational framework for the Bayesian models, allowing for fast and accurate approximation of posterior distributions for model parameters and latent effects.

Spatial and temporal variation in the data were addressed in the model by including terms for district (spatial resolution) and week (temporal resolution), representing the locations and time periods during which data were collected. To capture unmeasured regional differences and temporal trends that could influence transmission, spatiotemporal random effects were incorporated. These random effects account for dependencies and variations not explained by observed covariates. Specifically, the model included two spatiotemporal random effects: $r_t$ (temporal) and $b_i$ (spatial), as well as a fixed effect v $v_{i,t}$ to represent other known drivers. First, for the expectation of $\Delta R_t$ within each city $i$, we constructed a base model below which can be expressed as:

$$\mu_{i,t} = 1 + r_t + b_i + v_{i,t}$$

Where: $r_t$ is a random walk model of order 1 (rw1) ($\Delta r_t = r_t - r_{t-1} \sim N\left(0, \tau^{-1}\right)$) which is used to account for temporal trends in the data over time; $b_i$ is a modified Besag-York-Mollie (BYM2) model which accounts for spatial variation across districts, capturing unobserved differences between regions that may influence transmission; and $v_{i,t}$ is a fixed effect representing the cumulative infection rate within the population, included to account for the potential impact of herd immunity acquired by natural infection in previous waves before mass vaccination. Second, as the evolution of COVID-19 is a complex process, and factors mentioned above might not be the only explanatory variables for the observed changes in transmission, we further examined the duration of public holidays as a fixed effect in models.

All covariates obtained at daily temporal resolution were averaged by week. To account for multicollinearity of factors, we calculated pair-wise Pearson correlations for these variables and the variance inflation factor (VIF) for candidate variables in linear regressions for the whole country (S11 Fig & S36 Fig). In order to account for any non-normally distributed data, we also calculated Kendall rank coefficients between explanatory variables in our model as a non-parametric exploration of multicollinearity (S12 Fig & S37 Fig). Estimations of multicollinearity were broadly similar using Pearson and Kendall rank correlation coefficients, with weaker associations found using Kendall rank coefficients. Collinear variables were therefore excluded based on the more conservative Pearson correlation coefficients. Variables excluded from further analyses, based on highest VIF score and Pearson correlation coefficients of 0.5, included relative humidity and UV radiation. Only variables with a VIF score of less than 2.5 were retained. The relative impacts of remaining factors was thus defined as the contributed percentage change in $\Delta R_t$.

We built models of increasing complexity by systematically incorporating combinations of mobility, temperature, precipitation, stringency of intervention policy and public holidays covariates into our base model. Model goodness of fit was assessed using the DIC and logarithmic score (logscore), consistent with previous studies [75], and final models for each pandemic wave were selected. DIC balances model accuracy against complexity by estimating the number of effective parameters, while the logarithmic scores measure the predictive power of the model when excluding one data point at a time, with smaller values for each denoting better fitting models.

Third, we used the distributed lag non-linear models (DLNMs) formulation by defining lagged model covariates and a cross-basis matrix and incorporating the resulting cross-basis functions into our Bayesian spatiotemporal modelling framework. Using this approach we explored exposure-lag response associations between the ratio of increase of $R_t$ in COVID-19 transmission, and changes in mobility, meteorological variations, and Stringency Index of intervention policy. DLNMs are a family of models that describe the lagged relationship between exposure and response variables in a model across both spatial and temporal dimensions [76]. DLNM models incorporate cross-basis functions that combine a lag-response

function of variables at the temporal dimension and an exposure-response function to present the potential non-linear relationship along with the change of one factor. The resulting bi-dimensional exposure-lag-response function  flexibly estimates the intensity of factors at varying time-lags after exposure [76].

Given the common delays from infection to diagnosis and reporting, and the delayed impact of NPIs on COVID-19 transmission, the lag-response impact of different factors on COVID-19 transmission were assessed by 0-3 weeks, with natural cubic splines selected for both the exposure and the lag dimensions, consistent with previously published literature [77]. Last, we tested 18 candidate models of increasing complexity (with regard to input variables and model structure) with DLNMs for the whole country, and rural, suburban, and urban areas, respectively (S2 Table). DLNM cross-basis functions were built using R packages 'dlnm' and 'splines' and model parameters were estimated using the INLA approach in R version 4.1.0 [68,78]. INLA approaches include a wide and flexible class of models ranging from generalized linear mixed models to spatial and spatiotemporal models that are less computationally intensive therefore avoiding problems with model convergence [73,78,79].

Finally, as no informed prior distribution estimates were available at the time of analyses, we explored the sensitivity of the best fit model to a range of uninformative priors. We specified a range of priors around the hyperparameters, i.e., $\tau$, $\theta_1$, and $\theta_2$, in our base model. Prior distributions were investigated for the best fit model using data for the Delta wave time period (S4 Table) and for the wave 1 time period (S7 Table) using the deviance information criterion (DIC). The choice of prior distributions applied to best fit models using data from both waves was found to elicit only negligible measurable differences in model hyperparameters and DIC. Therefore, the prior used in this study was a penalized complexity prior with the precision $t = 1/\sigma^2$, so that $\Pr(1/\sqrt{t} > 0.5) = 0.01$.

**Model performance and validation.**  Model goodness-of-fit was assessed using DIC scores to compare model performances and identify the best-fitting model for the whole country, and rural, suburban, and urban areas, respectively. We also calculated the difference in mean absolute error (MAE) between the baseline model and the final selected model for each pandemic wave in order to identify the proportion of districts in different regions of India for which a more complex data-driven model improved model fit. Cross-validations using a leave-one-week-out and leave-one-state-out approach were conducted to refit the selected model. This approach excluded one week or one state, respectively, from the fitting process during each cross-validation model iteration. Comparisons were made between observations and out-of-sample posterior predictive $R_t$ for each state and week of both pandemic waves investigated. In order to validate DLNM model results, based on the findings of lag-response associations from analyses above, we incorporated lag-adjusted covariates into our spatiotemporal Bayesian hierarchical modelling and compared results with observations obtained from Bayesian spatiotemporal models incorporating DLNM models built using cross-basis functions.

## Results

### Spatiotemporal heterogeneity of mobility changes in India during the pandemic

Compared with baseline mean mobility patterns during the first 8 weeks of 2020, domestic travel within India dropped dramatically after the COVID-19 pandemic was declared by the WHO and the country implemented its first lockdown for transmission containment (Fig 1). The lowest mobility level for domestic travel (26.9% of the pre-pandemic mean level) was observed at week 15 of 2020 (April 5 – 11, 2020). In June 2020, restrictions on opening shopping centres, religious places, hotels, and restaurants were lifted [28], coinciding with

increased population flows and an increase in infection cases. Overall, mobility gradually recovered from mid-May 2020 to early March 2021, even during the first wave of COVID-19 in the second half of 2020.

A second lockdown was implemented across the country from mid-April to early June 2021 following a surge in transmission in March 2021 and concern about increased infections and deaths caused by the Delta variant. Domestic mobility during the second lockdown reduced significantly from an average level of 90.5% in the 8 weeks between January 31 – March 27, 2021, reaching its lowest level (54.6%) at week 20 of 2021 (May 16 – 22). However, the stringency, compliance and duration of mobility reductions were less strict and shorter than those of the first wave. Changes in mobility between rural, suburban and urban districts of India displayed similar temporal patterns (**Fig 1C**), but travel in urban areas (73.9%) was more affected by the pandemic compared with mobility in semi-urban (91.7%) and rural (94.4%) areas in 2020–2021 (**Fig 2**).

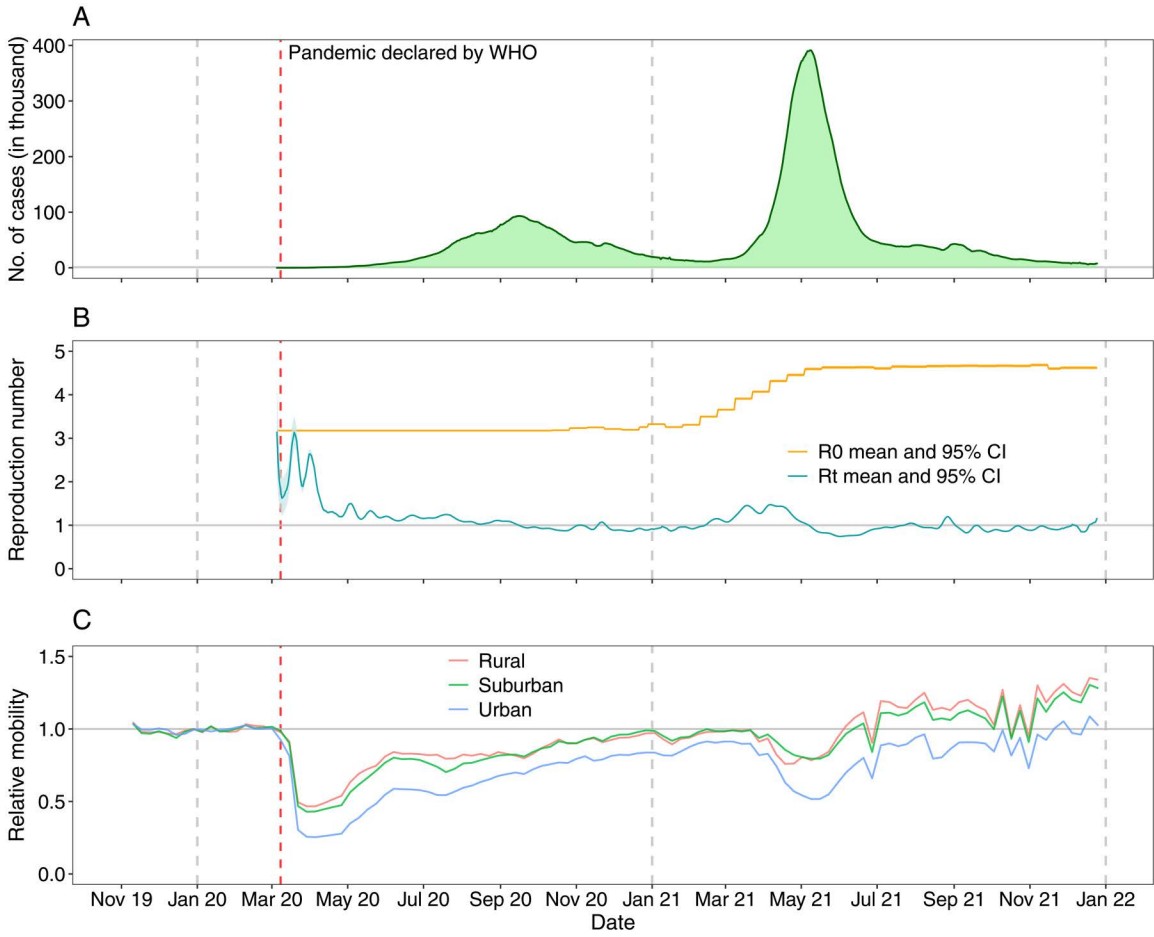

**Fig 1. COVID-19 cases, reproduction numbers and mobility changes in India during the pandemic. (A)** Number of daily new confirmed COVID-19 cases reported in India from March 15, 2020, to December 25, 2021. **(B)** Estimated mean and 95% confidence interval (CI) of the basic reproduction number ( $R_0$ ) and instantaneous reproduction rate ( $R_t$ ). **(C)** Relative weekly mobility of domestic travel by rural, suburban and urban areas in India as measured by the aggregated Google COVID-19 mobility research dataset. Relative mobility levels were standardized by the overall mean level of each type of flow in each region during the first 8 weeks of 2020. The red and grey vertical dashed lines indicate the date of the COVID-19 pandemic being declared by the WHO and the first date of each year, respectively.

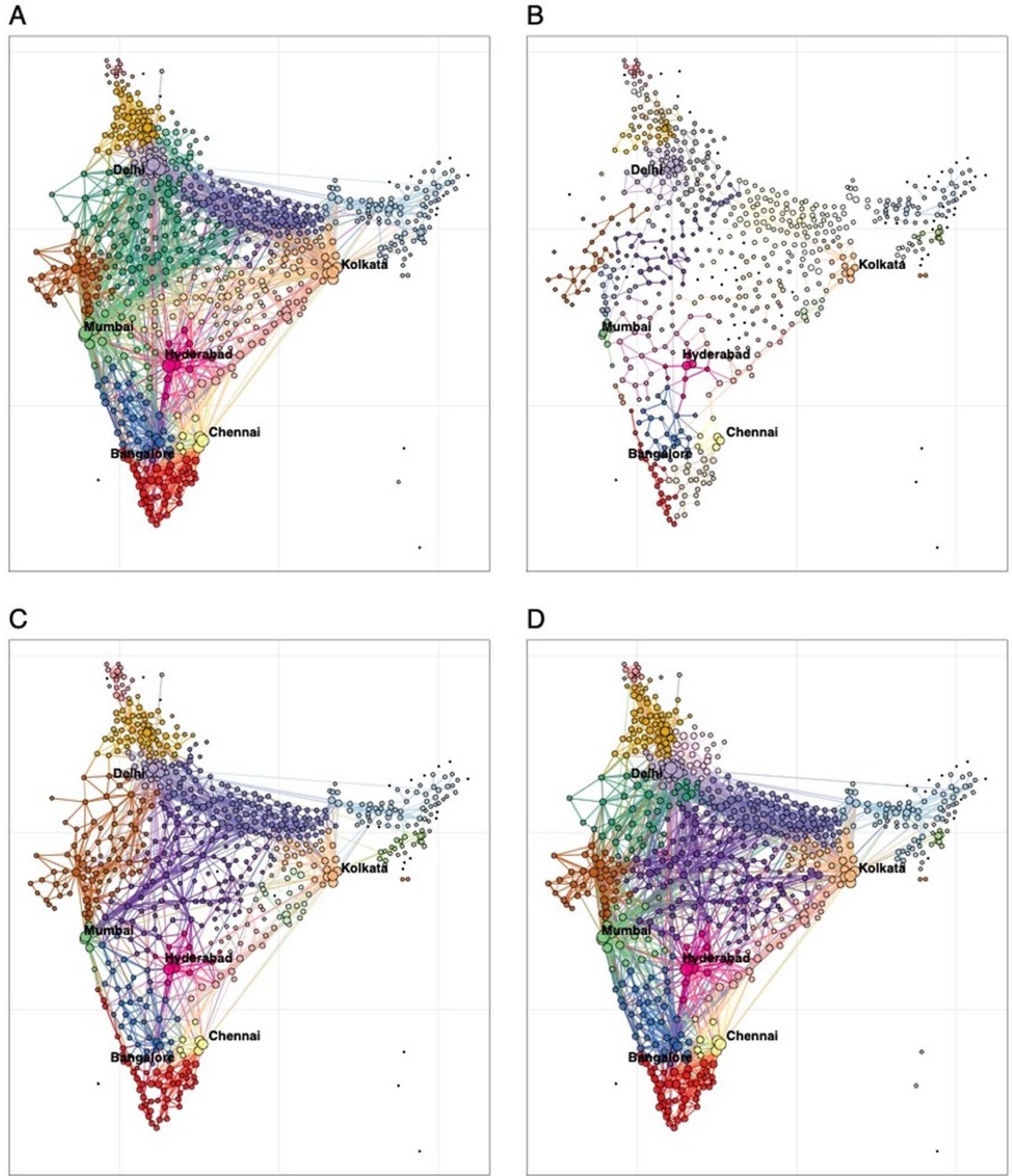

**Fig 2. Changes in community domestic travel networks of Indian districts across four time periods in 2019-2021. (A)** Communities (n=23) of domestic travel at district level during the pre-pandemic period from November 10, 2019, to February 22, 2020. **(B)** Communities (n=79) of domestic travel during the first lockdown on March 22 - May 2, 2020. **(C)** Communities (n=31) of domestic travel during the second lockdown on April 18 - May 29, 2021. **(D)**

Communities (n=22) of domestic travel post-second lockdown period (8 weeks), from November 7 to December 31, 2021, after travel restrictions for COVID-19 had been lifted in India. In each panel, geographically adjacent areas of the same colour represent an internally and closely connected community in terms of human movement in India. The community structure was detected using the Louvain algorithm, based on the aggregated Google COVID-19 mobility research dataset. Circle size represents the relative volume of outbound travellers. The bigger the circle, the higher the level of outflow.

Geographic groupings of connected districts also exhibited spatiotemporal heterogeneity in response to mitigation efforts. During the pre-pandemic period, districts formed 23 connected travel communities, with 13 major communities encompassing 94.4% of all districts. Connections between districts were severely disrupted during the first lockdown, forming 79 isolated communities, with 54.4% consisting of a single district (**S3A Fig**). In contrast, the second lockdown resulted in 31 communities, with 13 major communities covering 93.8% of districts—closer to pre-pandemic patterns. By late 2021, district connections had largely returned to their pre-pandemic state (**S3D Fig**).

## Nonlinear and lag-response impacts of mobility and other factors on the Delta wave

Bayesian spatiotemporal models with distributed lag nonlinear models (DLNMs) identified key lagged drivers of COVID-19 transmission during the Delta wave. Initial analysis excluded population density, humidity, and UV radiation due to multicollinearity or lack of significance (**S2 Table**). The inclusion of DLNMs for mobility, temperature, precipitation, and the Stringency Index (Model 4.1), including the holiday variable as a fixed effect in different candidate models and lagged between 0 and 3 weeks, resulted in a greater reduction in the DIC and mean logarithmic score compared with the baseline model (**S18 Fig**). For semi-urban areas, models which included DLNMs for mobility, temperature, and Stringency Index (Model 3.1) had the smallest DIC and logarithmic score, while for rural areas, only the Stringency Index was significant due to smaller mobility reductions.

Recovery of mobility to 99% of pre-pandemic levels and a Stringency Index below 68 significantly increased the Rt of COVID-19 transmission in India during the study period (**Fig 3A** & **3J**). An increase in weekly precipitation (>0.15m) and cooler weather (<27.2°C) also increased transmission risk, though the effects of extreme cold (<0°C) were not significant (**Fig 3D** & **3G**). Lagged impacts were apparent, with maximum effects observed at 1–2 weeks for mobility reductions and intervention policies. Similar lag-response patterns were found across urban, suburban, and rural areas, though the timing and magnitude of associations with the Stringency Index varied by region (**Fig 4**). Posterior predictive results from the best fitting model by cross-validation showed that the model had a robust performance compared to observed data (**S13**-**S16 Fig**). Spatial random effects and the fitted Rt for the whole country are also presented in the Supplementary Information (**S17**-**S20 Fig**).

Given the reporting delays of cases after exposure (i.e., incubation period plus the lags from illness onset, diagnosis to reporting, normally 10 days with an interquartile range of 8 – 11 days [80], we found that the introduction of DLNMs improved model adequacy statistics compared with the inclusion of factors with no lags, which confirmed the rationale and necessity of considering the lag-response effects in the modelling. The maximum associations of mobility reductions (**Fig 3B**; relative mobility >0.5 times baseline mobility associated with RR of <0.8)) and intervention policy (**Fig 3K**; Stringency Index <30 associated with RR <0.95) with changes in $R_t$ of COVID-19 transmission were found at a lag of 2 weeks with precipitation having an apparent maximum impact at a 1 to 2-week lag. However, we also found an increasing/decreasing risk of transmission under cool/hot weather at one 1-week lag (**Fig 3F**; temperature of <20°C associated with RR >1). Similar lag-response patterns between

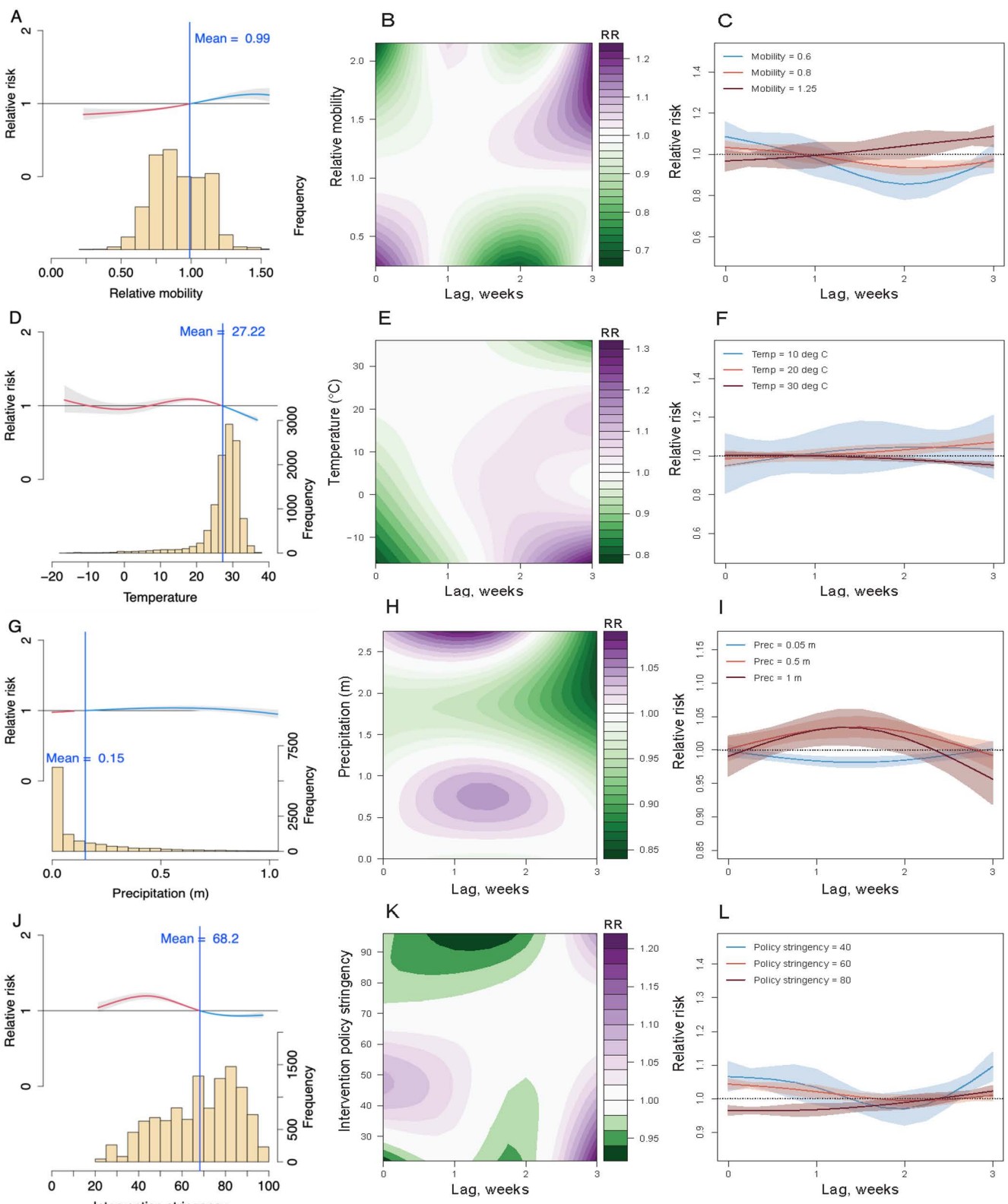

**Fig 3. The lagged impact of different factors and scenarios on COVID-19 transmission during the Delta wave in 2021. (A)** The overall association between mobility changes and COVID-19 transmission dynamics under 0- to 3-week lags. The red/blue lines show ratio of Rt under the scenario of mobility below/above the overall mean level (0.99). The histogram with the secondary y-axis shows the frequency of data under different levels. **(B)** Contour plot of

the association between mobility and risk of COVID-19 transmission. The deeper the shade of purple, the greater the increase in transmission risk, while the deeper the shade of green, the greater the decrease in Rt. **(C)** COVID-19 lag–response association for mobility level at 0.6, 0.8, 1.25, relative to the overall pre-pandemic mean level (1). The mean and 95% CI were presented. **(D)** – **(F)** Lag-response association between COVID-19 transmission and temperature (Temp) for cool (10°C), warm (20°C), and hot (30°C) weather, relative to the overall mean of 27.2°C. **(G)** – **(I)** COVID-19 lag-response association for precipitation (Prec) at 0.05, 0.5, and 1m, relative to the overall mean of 0.15m. **(J)** – **(L)** Lag-response association between COVID-19 transmission and the stringency of intervention policy at low (40), medium (60), and high (80), relative to the overall mean Stringency Index (68.2). Results are for the best fitting model with DLNMs (base model + mobility + temperature + precipitation + intervention policy; see SI Table S2) across the whole country.

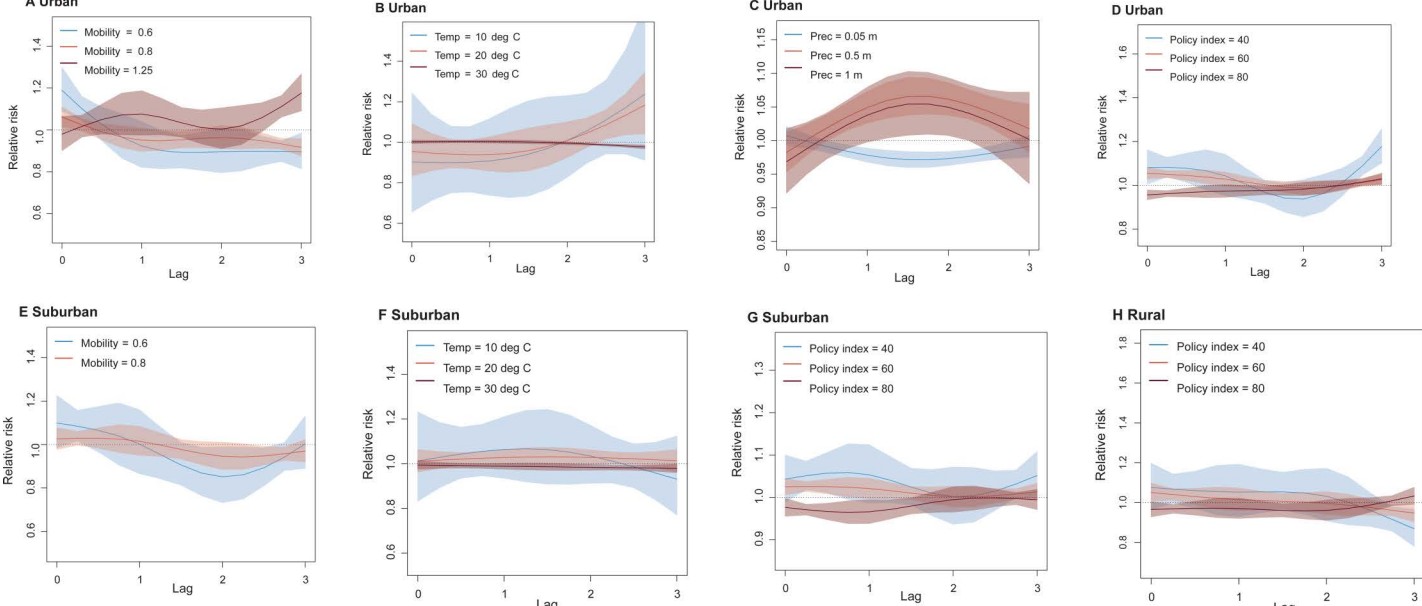

**Fig 4. The lag-response association between COVID-19 transmission and different factors in urban, suburban, and rural districts. (A)** – **(D)** COVID-19 lag–response association for different levels of mobility, temperature (Temp), precipitation (Prec) and the stringency of intervention policy in urban areas, relative to the overall mean level. Results are for the best fitting model with DLNMs (base model + mobility + temperature + precipitation + intervention policy) in urban districts. **(E)** – **(G)** Lag–response association between the risk of COVID-19 transmission and different levels of mobility, temperature (Temp), and the stringency of intervention policy in semi-urban areas, based on the best fitting model with DLNMs (base model + mobility + temperature + intervention policy; see SI Table S2) in suburban districts. **(H)** COVID-19 lag–response association for the Stringency Index of intervention policy at low (40), medium (60), and high (80), based on the best fitting model with DLNMs (base model + intervention policy) in rural districts. The mean and 95% CI of RR for each level were presented.

COVID-19 transmission and covariates at different levels were also found in urban, suburban, and rural districts (**Fig 4**). The results from leave-one-week-out cross-validation showed the best fitting 2-week lag-response model could further improve the prediction of dynamics in Rt of COVID-19 transmission across India (**S23**-**S27 Fig**).

## Comparing lag-response impacts of different factors between waves

We also ran Bayesian spatiotemporal models with DLNMs using data from 19th July to 29th November 2020 to compare drivers of transmission during both pandemic waves in 2020 (initial transmission wave) and 2021 (Delta wave). Results of DLNMs exploring drivers of COVID-19 transmission during the first wave were consistent with those exploring associations of COVID-19 transmission during the Delta wave in India. A rebound in mobility to between 1.2 and 1.4 times the mobility of pre-pandemic levels resulted in an increase in RR (>1) with a lag-time of between one and two weeks (**Fig 5C**) and high Stringency Index (80) was associated with a lower RR with a two-and-a-half-week lag (**Fig 5L**).

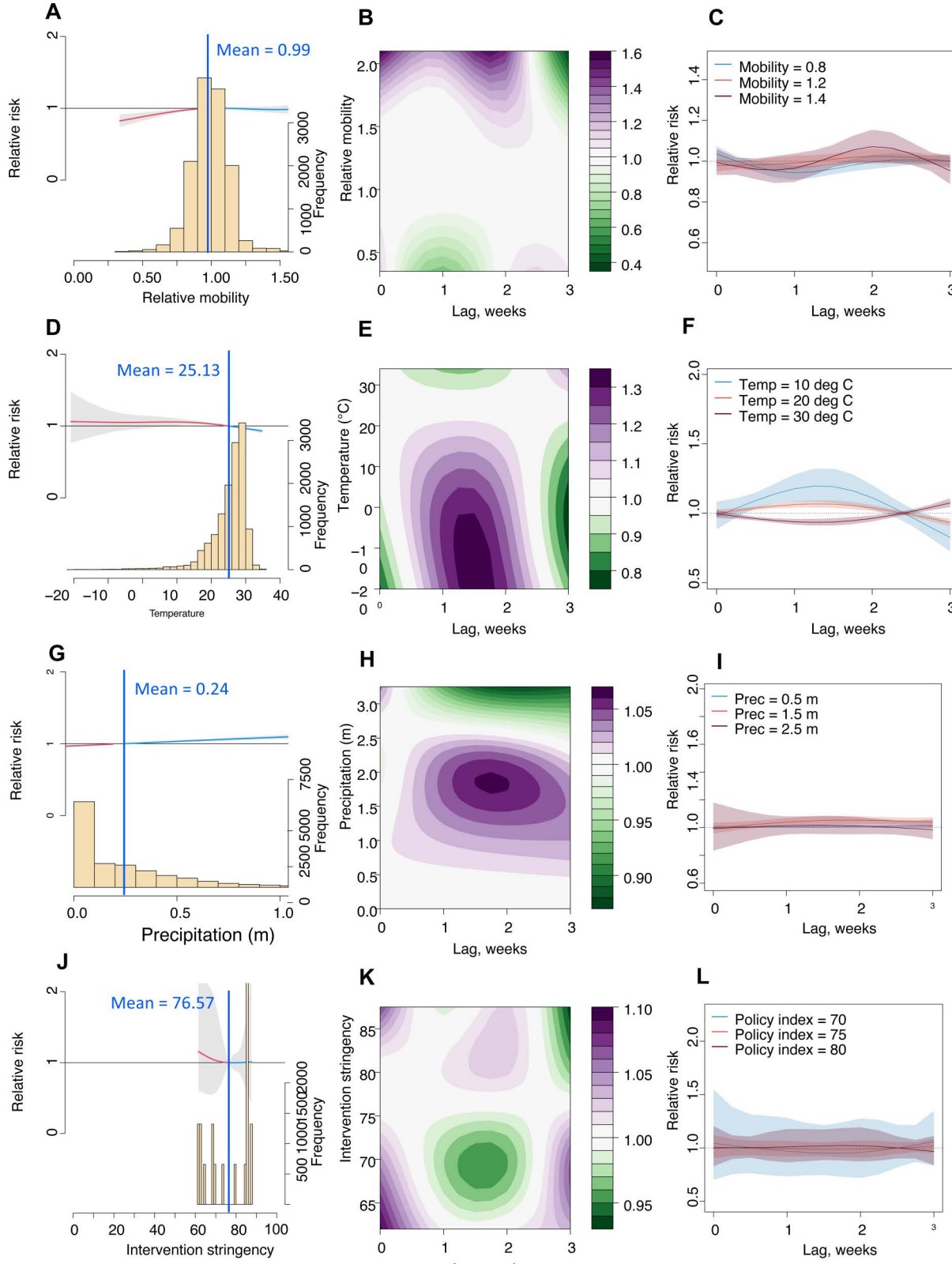

**Fig 5. The lagged impact of different factors and scenarios on COVID-19 transmission during the initial wave of COVID-19 transmission in the second half of 2020. (A)** The overall association between mobility changes and COVID-19 transmission dynamics under 0- to 3-week lags. The red/blue lines show RR under the scenario of mobility below/above the overall mean level (0.99). The histogram with

the secondary y-axis shows the frequency of data under different levels. (B) Contour plot of the association between mobility and relative risk (RR) of COVID-19 transmission. The deeper the shade of purple, the greater the increase in RR of transmission, while the deeper the shade of green, the greater the decrease in RR. (C) COVID-19 lag–response association for mobility level at 0.8, 1.2, 1.4 relative to the overall pre-pandemic mean level (1). The mean and 95% CI were presented. (D) – (F) Lag-response association between COVID-19 transmission and temperature (Temp) for cool (10°C), warm (20°C), and hot (30°C) weather, relative to the overall mean of 25°C. (G) – (I) COVID-19 lag-response association for precipitation (Prec) at 0.5, 1.5, and 2.5m, relative to the overall mean of 0.24m. (J) – (L) Lag-response association between COVID-19 transmission and the stringency of intervention policy at three different measures of stringency: 70, 75, and 80, relative to the overall mean Stringency Index (76.5). Results are for the best fitting model with DLNMs (base model + mobility + temperature + precipitation + intervention policy; see SI Table S2) across the whole country.

Based on best fit model statistics (lowest DIC and mean logarithmic score compared to baseline model) DLNMs which best fit the data were models which included mobility, temperature, precipitation and Stringency Index (Model 4.1; **S3 Table**). Cold weather (10°C & 20°C) was associated with a higher RR (**Fig 5D** & **5F**) and an increase in weekly precipitation (>0.2m) was also associated with an increase in transmission risk (**Fig 5G**) with a lag-time increase of between 1 and 2 weeks. Consistent with model validation using data for the Delta wave, cross-validation showed robust model results when posterior predictive results for the initial wave in 2020 were compared with observed data (**S38** – **S41 Fig**).

## Discussion

Using a de-identified and aggregated Google COVID-19 mobility research dataset, derived from time- and space-explicit mobile phone data, our study quantified changes in population movements across rural and urban districts, and identified connected communities of travel networks, in India over the course of the pandemic. Our modelling results showed that mobility changes, together with stringency of government interventions and climate factors had lagged-response impacts on the risk of COVID-19 transmission. The first nationwide lockdown between March and June 2020, together with a reduction in population mobility, appear to have been the main drivers of a relatively low transmission wave of COVID-19 in India during the first half of 2020 [21,81]. Although the announcement of the lockdown had initially resulted in an increase in population mobility, with workers mostly representing informal sectors travelling interstate to return home [82], the majority of people travelling were not infected and this population mobility therefore had little impact on transmission [5].

In early 2021, NPI restriction measures such as social distancing and mask-wearing had been gradually eased due to a sense of COVID-19 clinical infections being mild [8,83], and inter-state and rural to urban human mobility was seen to be increasing [21,83]. This included mass attendance of political rallies and religious festivals, such as the Hindu festival Kumbh Mela in India's most populous state of Uttar Pradesh where hundreds of thousands of people gathered at the banks of the River Ganges [21,23,83]. The modelling results presented here indicate that this recovery of mobility in early 2021 to 99% that of pre-pandemic levels, together with lower stringency of government interventions and emergence of the more transmissible Delta variant, contributed to higher transmission of COVID-19 infection during the Delta pandemic wave. This is consistent with previously published research which attributed the surge of COVID-19 in April 2021 to the emergence of the more transmissible Delta variant (B.1.617 lineage) and dominance as the main circulating strain, as well as relaxation of NPIs [13,21,84].

The second lockdown with reduced travel frequency and contact rates among populations also played a significant role in mitigating COVID-19 spread across districts and transmission in communities in the country. Mobility patterns were inversely associated with the national Stringency Index, with a relative drop in mobility below 50% associated with a Stringency

Index of 80, consistent with previous research which found that community mobility, based on Google location data, drastically fell after the lockdown was implemented. . However, the impacts of mobility changes were not fully synchronized between rural and urban areas, and the effects of travel restrictions and other interventions in slowing down COVID-19 transmission hinged on the intensity of these measures in reducing $R_t$ of new variants with a higher transmissibility. Model results showed differences in lagged associations of COVID-19 RR with Stringency Index between rural, semi-urban and urban districts [24] and this was reflected in urban vs. rural transmission dynamics between both pandemic waves. During the first wave of COVID-19 in India, transmission was higher in urban rather than rural settings and cases were spatially clustered throughout metropolitan and peri-urban areas [33,85,86]. Conversely, during the Delta pandemic wave in India, cases were observed to be spreading more in rural areas where access to healthcare can be more limited than in urban areas [18].

Previous research had also observed significant associations between number of COVID-19 cases and temperature, dew point, humidity and solar radiation [28,34,54,65,87]. Our Bayesian spatiotemporal model results were consistent with these findings observing lag-response associations between COVID-19 transmission and climate covariates (temperature and precipitation), although these effects appear to be very limited in terms of relative risk. Model results for the Delta pandemic wave found that a decrease in temperature (<20°C) was associated with an increased relative risk, consistent with previous modelling studies exploring climate impacts on COVID-19 transmission in India [32], and an increase in precipitation (>2.5m) associated with a decreased relative risk, with a 1 to 2-week lagged impact. This is consistent with wave 1 modelling results which found a 1 to 2-week lagged association between cold weather and precipitation on an increase in RR of COVID-19 transmission.

The work we have presented builds upon previous research exploring the driving factors that led to the surge in COVID-19 transmission during the Delta pandemic wave in India [15,23,27], while also presenting a number of novel factors not previously presented in the literature. Firstly, to our knowledge this is the first study to explore inter-district mobility patterns in India during the initial and Delta waves of COVID-19 transmission, relative to pre-pandemic levels, delineated by urban, suburban and rural location. By investigating these changes in human mobility using fine spatial resolution Google COVID-19 Aggregated Mobility Research data we have demonstrated that a surge in population movement, together with an easing of NPIs were the main contributors to the surge in transmission during the Delta pandemic wave. To our knowledge, this is also the first study to combine human mobility data with Stringency Index and climate data within a Bayesian spatiotemporal framework to compare drivers of transmission by urban, suburban and rural district over the course of the pandemic in India, and quantify the lagged impact of these drivers on COVID-19 transmission risk.

Our modelling approach explored the spatiotemporal heterogeneities in drivers of transmission at district level accounting for urbanisation, building upon previous research exploring the association between state level urbanisation and COVID-19 transmission [29]. Using a Bayesian spatiotemporal framework that incorporates spatial and temporal dependencies into models is particularly useful in regions such as India with substantial divergence between urban and rural areas [88,89]. Additionally, Bayesian hierarchical models provide flexibility for quantifying heterogeneities in spatiotemporal drivers of transmission during both pandemic waves while allowing complex and nonlinear relationships within the data to be captured [78]. Building upon this framework by integrating novel DLNM models into the Bayesian framework allowed us to quantify lagged, nonlinear associations of drivers of transmission with COVID-19 incidence, which is critical for quantifying the lagged impact of interventions on transmission and to account for the infection incubation period and delays in case reporting.

While our findings represent a comprehensive understanding of the drivers of transmission during the initial and Delta waves of COVID-19 transmission in India, these results should be interpreted in light of several important limitations. First, the Google mobility data is limited to smartphone users who have opted into Google's Location History feature, which is off by default. These data may not be representative of the population as whole, and furthermore their representativeness may vary by location. Importantly, these limited data are only viewed through the lens of differential privacy algorithms, specifically designed to protect user anonymity and obscure fine detail. However, comparisons between mobility datasets have shown good agreement with Google Location History data and other commonly used mobility data sources for capturing population-level mobility patterns [90]. Moreover, comparisons across rather than within locations are only descriptive since these regions can differ in substantial ways.

Second, the accuracy of our models relied on accurate estimates of $R_t$ derived from reported case data, with $R_0$ estimates proportional to the contact rate, and might vary according to the local situation. The quality of reported data likely differed across districts due to varying case definitions, testing and surveillance capacity across the country, with various underreporting rates and reporting delays. Third, the Stringency Index data at state level used in spatiotemporal analyses for districts was formulated to assess lockdown strictness and measure the political commitment and strictness of governmental policies. These data did not measure the effectiveness of a country's response or provide information on how well policies were enforced. A higher value of Stringency Index did not necessarily mean that a country's response was better than that of those with lower values [24,54]. Fourth, many other factors (e.g., vaccination and prior infections) might also contribute to COVID-19 transmission, but our models did not specify the contributions of these factors.

The model results and modelling approach we have described here are critical to our understanding of drivers of COVID-19 in India, and elsewhere, and makes an important contribution to our understanding of human mobility, NPIs and climate drivers on infection transmission. As stated previously, while COVID-19 is no longer a public health emergency, it remains a pandemic with substantial associated long-term health impacts and mortality. Understanding the lagged impact of human mobility, climate and interventions on infection transmission, and heterogeneity in drivers between urban and rural settings, is crucial for predicting seasonal transmission dynamics and for allocating resources such as mass testing and vaccination campaigns. The Bayesian spatiotemporal framework incorporating DLNMs we have presented provides a valuable framework for understanding the impact of drivers of COVID-19 transmission, and for understanding future novel infections which emerge due to our urbanised global society, with more extreme weather events and pronounced changes in climate [91,92].

## Supporting information

**S1 Fig. Regions in India investigated by this study and the number and density of population at district level (administrative level II) in 2020.** Areas shaded in grey are areas for which no data is available. **S2 Fig.** Five periods for travel network modularity analysis (A): 1) Pre-pandemic period (15 weeks) from November 10, 2019 to February 22, 2020; 2) First lockdown (6 weeks), from March 22 to May 2, 2020, that included strict travel restrictions, stay-at home orders and closure of many businesses; 3) Pre-second lockdown period (8 weeks) from January 31 to March 27, 2021; 4) Second lockdown (6 weeks) for the Delta wave, from April 18 to May 29, 2021; 5) post-second lockdown period (8 weeks), from November 7 to December 31, 2021, after travel restrictions for COVID-19 had been lifted in India. **S3 Fig.**

Relative changes of outbound travel from districts across India during the pandemic compared with average pre-pandemic levels during the 12 weeks from November 10, 2019, to February 22, 2020. (**A**) Reductions of outbound flows under the first lockdown during the 6-week period from March 22 to May 2, 2020. (**B**) Changes in outflow during the 8-week period from January 31 to March 27, 2021, before the second lockdown. (**C**) Reductions of outflows during the 6-week second lockdown from April 18 to May 29, 2021. (**D**) Changes in outflow during the 8-week period from November 7 to December 31, 2021. Sub-division maps at administrative level I (state) and II (district) were obtained from the GADM version 3.6 (https://gadm.org/). Regions in which outflow data are not available are those represented in green. Areas shaded in grey are areas for which no data is available. **S1 Table**. Summary Statistics for data used for wave 1 and Delta wave spatiotemporal models **S2 Table**. Wave 2: Adequacy results for models with DLNMs and increasing complexity. **S3 Table**. Wave 2: Adequacy results for models (without DLNMs) using 2-week lag covariates with increasing complexity. **S4 Table**. Model hyperparameters using a range of prior distributions in best fit model 4.1 for Delta Wave. **S4 Fig** Relative intra-district mobility during the Delta wave in India, standardised by pre-pandemic mean baseline levels of mobility for the first eight weeks of 2020 (December 29, 2019 – February 22, 2020) for each district. The weeks in 2021 investigated are numbered in maps. Areas shaded in grey are areas for which no data is available. **S5 Fig**. Stringency Index of COVID-19 intervention policy implemented during the Delta wave in India. The weeks in 2021 investigated are numbered in maps. Areas shaded in grey are areas for which no data is available. **S6 Fig**. Mean temperature at 2m above the surface during the Delta wave in India. The weeks in 2021 investigated are numbered in maps. Areas shaded in grey are areas for which no data is available. **S7 Fig**. Accumulated weekly precipitation (metres) during the Delta wave in India. The weeks in 2021 investigated are numbered in maps. Areas shaded in grey are areas for which no data is available. **S8 Fig**. Relative humidity during the Delta wave in India. The weeks in 2021 investigated are numbered in maps. Areas shaded in grey are areas for which no data is available. **S9 Fig**. Downward ultraviolet (UV) radiation (KJ/m2 per hour) during the Delta wave in India. The weeks in 2021 investigated are numbered in maps. Areas shaded in grey are areas for which no data is available. **S10 Fig**. Weekly Rt derived from COVID-19 cases reported during the Delta wave in India. The weeks in 2021 investigated are numbered in maps. Areas shaded in grey are areas for which no data is available. **S11 Fig**. Pairwise Pearson correlations between weekly means of variables at district level during the Delta wave in India, 2021. R0: basic reproduction number. Rt: instantaneous reproduction number. ln_R: log(Rt/R0). Cases_rate: new COVID-19 cases reported per 1000 people. Cases_accu_rate: cumulative cases per 1000 people reported since the first week of the wave. mean_intra: intra-district relative mobility. d2m: relative humidity. t2m: mean temperature of air (°C at 2m above the surface of land, sea or inland waters). tp: precipitation (metres). uv: downward ultraviolet radiation. Stringency: index of COVID-19 intervention stringency. Holiday: days of public holidays in a week. pop_sum: total population of each district. pop_density: population number per km2 of each district. **S12 Fig**. Kendall rank correlations between weekly means of variables at district level during the Delta wave in India, 2021. R0: basic reproduction number. Rt: instantaneous reproduction number. ln_R: log(Rt/R0). Cases_rate: new COVID-19 cases reported per 1000 people. Cases_accu_rate: cumulative cases per 1000 people reported since the first week of the wave. mean_intra: intra-district relative mobility. d2m: relative humidity. t2m: mean temperature of air (°C at 2m above the surface of land, sea or inland waters). tp: precipitation (metres). uv: downward ultraviolet radiation. Stringency: index of COVID-19 intervention stringency. Holiday: days of public holidays in a week. pop_sum: total population of each district. pop_density: population number per km2 of each district. **S13 Fig**. Posterior predictive mean Rt during the Delta wave

in India, 2021, derived from the best fitting model (model 4.1) at country level using leave-one-week-out cross-validation approach. The weeks in 2021 investigated are numbered in maps. Areas shaded in grey are areas for which no data is available. **S14 Fig**. Standard deviation (SD) of posterior predictive Rt during the Delta wave in India, 2021, derived from the best fitting model (model 4.1 without DLNMs) at country level using a leave-one-week-out cross-validation approach. Areas shaded in grey are areas for which no data is available. **S15 Fig**. Posterior predictive mean Rt during the Delta wave in India, 2021, derived from the best fitting model (model 4.1) at country level using leave-one-state-out cross-validation approach. The weeks in 2021 investigated are numbered in maps. Areas shaded in grey are areas for which no data is available. **S16 Fig**. Standard deviation (SD) of posterior predictive Rt during the Delta wave in India, 2021, derived from the best fitting model (model 4.1 without DLNMs) at country level using a leave-one-state-out cross-validation approach. Areas shaded in grey are areas for which no data is available. **S17 Fig**. Contribution of spatial random effects to estimates of Rt changes in the base model. Areas shaded in grey are areas for which no data is available. **S18 Fig**. Improvement by using the best fitting model across the country, compared to baseline model. Difference between mean absolute error (MAE) for the baseline model (weekly random effects, spatial random effects and population density) and MAE for the best fitting model (model 4.1 with DLNMs). Districts with positive values (pink) suggest that capturing the nonlinear and delayed impacts of mobility, climate information and intervention stringency, improves the model in these areas. Districts with negative values (blue) suggest that mobility, intervention and climate information did not improve the model fit and other unexplained factors might dominate space-time dynamics in these areas. The MAE of the selected model was smaller than the baseline model for 385 of the 665 (57.9%) districts in India, with the results of model performance provided by geo-political regions in the Table. Areas shaded in grey are areas for which no data is available. **S19 Fig**. Observed versus posterior fitted Rt in the capital district of each state using the best fitting model (model 4.1 with DLNMs) at country level. Graphs with a log scale at y-axis show the observed Rt derived from reported case data, and corresponding mean and 95% confidence interval (CI, shaded pink area) of fitted Rt, derived from the best fitting model (model 4.1 with DLNMs) at country level. States are ordered by their geographical location. **S20 Fig**. Observed versus posterior predictive Rt in the capital district of each state, using leave-one-week-out cross-validation approach. Graphs with a log scale at y-axis show the observed Rt derived from reported case data, and corresponding posterior predictive mean and 95% prediction interval (CI, shaded pink area), derived from the best fitting model (model 4.1 with DLNMs) at country level. States are ordered by their geographical location. **S21 Fig**. Contribution of spatial random effects to estimates of Rt changes in the base model. Areas shaded in grey are areas for which no data is available. **S22 Fig**. Improvement of using the best fitting model with 2-week lag covariates (no DLNMs), compared to baseline model with the same lag. Difference between mean absolute error (MAE) for the baseline model and MAE for the best fitting model (Model 4.1). Districts with positive values (pink) suggest that capturing the 2-week lag impacts of mobility, temperature, UV and intervention stringency, improves the model in these areas. Districts with negative values (blue) suggest that mobility, intervention and climate information did not improve the model fit and other unexplained factors might dominate space-time dynamics in these areas. The MAE of the selected model was smaller than the baseline model for 428 of the 665 (64.4%) districts in India, and further improved the best fitting model with DLNMs (_Fig.tifS12). Results of model performance are provided by geo-political regions in the Table. Areas shaded in grey are areas for which no data is available. **S23 Fig**. Posterior predictive mean Rt during the Delta wave in India, 2021, derived from the best fitting model (model 4.1 without DLNMs) at country level using 2-week lag covariates

and leave-one-week-out cross-validation approach. Areas shaded in grey are areas for which no data is available. **S24 Fig**. Standard deviation (SD) of posterior predictive Rt during the Delta wave in India, 2021, derived from the best fitting model (model 4.1 without DLNMs) at country level using 2-week lag covariates and leave-one-week-out cross-validation approach. Areas shaded in grey are areas for which no data is available. **S25 Fig. Observed versus posterior predictive Rt in the capital district of each state.** Graphs with a log scale at y-axis show the observed Rt derived from reported case data, and corresponding posterior predictive mean and 95% prediction interval (CI, shaded pink area), derived from the best fitting model without DLNMs at country level (model 4.1: base model + mobility + temperature + UV + intervention policy; see SI Table S2), using 2-week lag covariates and leave-one-week-out cross-validation approach. States are ordered by their geographical location. **S26 Fig**. Posterior predictive mean Rt during the Delta wave in India, 2021, derived from the best fitting model (model 4.1 without DLNMs) at country level using 2-week lag covariates and leave-one-state-out cross-validation approach. Areas shaded in grey are areas for which no data is available. **S27 Fig**. Standard deviation (SD) of posterior predictive Rt during the Delta wave in India, 2021, derived from the best fitting model (model 4.1 without DLNMs) at country level using 2-week lag covariates and leave-one-state-out cross-validation approach. Areas shaded in grey are areas for which no data is available. **Table S5.** Wave 1: Adequacy results for models with DLNMs and increasing complexity. **Table S6.** Wave 1: Adequacy results for models (without DLNMs) using 2-week lag covariates with increasing complexity. **Table S7.** Model hyperparameters using a range of prior distributions in best fit model 4.1 for **S28 Fig**. COVID-19 cases reported by district each week during wave 1 in India. The weeks in 2020 investigated are numbered in maps. Areas shaded in grey are areas for which no data is available. **S29 Fig**. Relative intra-district mobility during wave 1 in India, standardised by pre-pandemic mean baseline levels of mobility for the first eight weeks of 2020 (December 29, 2019 – February 22, 2020) for each district. The weeks in 2020 investigated are numbered in maps. Areas shaded in grey are areas for which no data is available. **S30 Fig**. Stringency Index of COVID-19 intervention policy implemented during wave 1 in India. The weeks in 2020 investigated are numbered in maps. Areas shaded in grey are areas for which no data is available. **S31 Fig**. Mean temperature at 2m above the surface during wave 1 in India. The weeks in 2020 investigated are numbered in maps. Areas shaded in grey are areas for which no data is available. **S32 Fig**. Accumulated weekly precipitation (metres) during wave 1 in India. The weeks in 2020 investigated are numbered in maps. Areas shaded in grey are areas for which no data is available. **S33 Fig**. Relative humidity during wave 1 in India. The weeks in 2020 investigated are numbered in maps. Areas shaded in grey are areas for which no data is available. **S34 Fig**. Downward ultraviolet (UV) radiation (KJ/m2 per hour) during wave 1 in India. The weeks in 2020 investigated are numbered in maps. Areas shaded in grey are areas for which no data is available. **S35 Fig**. Weekly Rt derived from COVID-19 cases reported during the wave 1 in India. The weeks in 2020 investigated are numbered in maps. Areas shaded in grey are areas for which no data is available. **S36 Fig**. Pairwise Pearson correlations between weekly means of variables at district level during the wave 1 in India, 2020. R0: basic reproduction number. Rt: instantaneous reproduction number. ln_R: log(Rt/R0). Cases_rate: new COVID-19 cases reported per 1000 people. Cases_accu_rate: cumulative cases per 1000 people reported since the first week of the wave. mean_intra: intra-district relative mobility. d2m: relative humidity. t2m: mean temperature of air (°C at 2m above the surface of land, sea or inland waters). tp: precipitation (metres). uv: downward ultraviolet radiation. Stringency: index of COVID-19 intervention stringency. Holiday: days of public holidays in a week. pop_sum: total population of each district. pop_density: population number per km2 of each district. **S37 Fig**. Kendall rank correlations between weekly means of variables at district level

during the wave 1 in India, 2020. R0: basic reproduction number. Rt: instantaneous reproduction number. ln_R: log(Rt/R0). Cases_rate: new COVID-19 cases reported per 1000 people. Cases_accu_rate: cumulative cases per 1000 people reported since the first week of the wave. mean_intra: intra-district relative mobility. d2m: relative humidity. t2m: mean temperature of air (°C at 2m above the surface of land, sea or inland waters). tp: precipitation (metres). uv: downward ultraviolet radiation. Stringency: index of COVID-19 intervention stringency. Holiday: days of public holidays in a week. pop_sum: total population of each district. pop_density: population number per km2 of each district. **S38 Fig**. Posterior predictive mean Rt during wave 1 in India, 2020, derived from the best fitting model (model 4.1) at country level using leave-one-week-out cross-validation approach. The weeks in 2020 investigated are numbered in maps. Areas shaded in grey are areas for which no data is available. **S39 Fig**. Standard deviation (SD) of posterior predictive Rt during wave 1 in India, 2020, derived from the best fitting model (model 4.1) at country level leave-one-week-out cross-validation approach. Areas shaded in grey are areas for which no data is available. **S40 Fig**. Posterior predictive mean Rt during wave 1 in India, 2020, derived from the best fitting model (model 4.1) at country level using leave-one-district-out cross-validation approach. The weeks in 2020 investigated are numbered in maps. Areas shaded in grey are areas for which no data is available. **S41 Fig**. Standard deviation (SD) of posterior predictive Rt during wave 1 in India, 2020, derived from the best fitting model (model 4.1) at country level leave-one-district-out cross-validation approach. Areas shaded in grey are areas for which no data is available. **S42 Fig**. Contribution of spatial random effects to estimates of Rt changes in the base model. Areas shaded in grey are areas for which no data is available. **S43 Fig**. Improvement by using the best fitting model across the country, compared to baseline model. Difference between mean absolute error (MAE) for the baseline model (weekly random effects, spatial random effects and population density) and MAE for the best fitting model (model 4.1 with DLNMs). Districts with positive values (pink) suggest that capturing the nonlinear and delayed impacts of mobility, climate information and intervention stringency, improves the model in these areas. Districts with negative values (blue) suggest that mobility, intervention and climate information did not improve the model fit and other unexplained factors might dominate space-time dynamics in these areas. The MAE of the selected model was smaller than the baseline model for 430 of the 661 (65.17%) districts in India, with the results of model performance provided by geo-political regions in the Table. Areas shaded in grey are areas for which no data is available. **S44 Fig**. Observed versus posterior fitted Rt in the capital district of each state using the best fitting model (model 4.1 with DLNMs) at country level. Graphs with a log scale at y-axis show the observed Rt derived from reported case data, and corresponding mean and 95% confidence interval (CI, shaded pink area) of fitted Rt, derived from the best fitting model (model 4.1 with DLNMs) at country level. States are ordered by their geographical location. **S45 Fig**. Observed versus posterior predictive Rt in the capital district of each state, using leave-one-week-out cross-validation approach. Graphs with a log scale at y-axis show the observed Rt derived from reported case data, and corresponding posterior predictive mean and 95% prediction interval (CI, shaded pink area), derived from the best fitting model (model 4.1 with DLNMs) at country level. States are ordered by their geographical location. **S46 Fig**. Posterior predictive mean Rt during the wave 1 in India, 2020, derived from the best fitting model (model 4.1 without DLNMs) at country level using 2-week lag covariates and leave-one-week-out cross-validation approach. Areas shaded in grey are areas for which no data is available. **S47 Fig**. Standard deviation (SD) of posterior predictive Rt during wave 1 in India, 2020, derived from the best fitting model (model 4.1 without DLNMs) at country level using 2-week lag covariates and leave-one-week-out cross-validation approach. Areas shaded in grey are areas for which no data is available. **S48 Fig**. Posterior predictive mean Rt during

the wave 1 in India, 2020, derived from the best fitting model (model 4.1 without DLNMs) at country level using 2-week lag covariates and leave-one-district-out cross-validation approach. Areas shaded in grey are areas for which no data is available. **S49 Fig**. Standard deviation (SD) of posterior predictive Rt during wave 1 in India, 2020, derived from the best fitting model (model 4.1 without DLNMs) at country level using 2-week lag covariates and leave-one-district-out cross-validation approach. Areas shaded in grey are areas for which no data is available.
(DOCX)

## Acknowledgments

We thank the researchers and organisations who generated and publicly shared the mobility, epidemiological, intervention, sequencing data, and analysing code used in this research. We also thank Ms. Xilin Wu for collecting genomic data and sharing code for R0 calculation and Dr. Wenbin Zhang and Dr. Edson Utazi for commenting on the modelling framework. The corresponding authors had full access to all the data in the study and had final responsibility for the decision to submit for publication. The views expressed in this article are those of the authors and do not represent any official policy.

## Author contributions

**Conceptualization:** Eimear Cleary, Fatumah Atuhaire, Andrew J Tatem, Shengjie Lai.

**Data curation:** Eimear Cleary, Alessandro Sorichetta, Alexander Cunningham, Massimiliano Pasqui, Marcello Schiavina, Michele Melchiorri, Maksym Bondarenko, Harry E R Shepherd.

**Formal analysis:** Eimear Cleary, Fatumah Atuhaire, Shengjie Lai.

**Methodology:** Eimear Cleary, Fatumah Atuhaire, Andrew J Tatem, Shengjie Lai.

**Validation:** Eimear Cleary.

**Visualization:** Eimear Cleary.

**Writing – original draft:** Eimear Cleary, Shengjie Lai.

**Writing – review & editing:** Eimear Cleary, Fatumah Atuhaire, Alessandro Sorichetta, Nick Ruktanonchai, Cori Ruktanonchai, Alexander Cunningham, Sabu S Padmadas, Amy Wesolowski, Derek A T Cummings, Andrew J Tatem, Shengjie Lai.

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
