## [Decision Letter · Decision Letter 0]

11 Nov 2024

PGPH-D-24-01332

Comparing lagged impacts of mobility changes and environmental factors on COVID-19 waves in rural and urban India: a Bayesian spatiotemporal modelling study

Dear Dr. Cleary,

Thank you for submitting your manuscript to PLOS Global Public Health. After careful consideration, we feel that it has merit but does not fully meet PLOS Global Public Health’s publication criteria as it currently stands. Therefore, we invite you to submit a revised version of the manuscript that addresses the points raised during the review process.

We look forward to receiving your revised manuscript.

Kind regards,

Sheikh Taslim Ali, M.Sc., Ph.D.

Academic Editor

Additional Editor Comments (if provided):

Authors should account the following key issues along with the reviewers' comments while revising the manuscript.

- I suggest exclude 'Bayesian' term from the title itself. Otherwise, justify mentioning it in the title specifically. Some of the key epidemiological parameters are misled in the report, including "the instantaneous basic reproduction number ( 0) over time", need to clarify of such new measures in the study or revise the text accordingly.

- Provide the more details on the construction of such Bayesian spatiotemporal modelling framework. Authors may required to include more information in the main text in brief and in the supplementary material with more details.

- As highlighted by the reviewers, several sensitivity analyses should be carried in the revised version.

- Finally, the figures are of poor quality to review, make sure better figure quality in the revised version to follow.

Reviewers' comments:

Reviewer's Responses to Questions

**Comments to the Author**

1. Does this manuscript meet PLOS Global Public Health’s publication criteria ? Is the manuscript technically sound, and do the data support the conclusions? The manuscript must describe methodologically and ethically rigorous research with conclusions that are appropriately drawn based on the data presented.

Reviewer #1: Yes

Reviewer #2: Yes

Reviewer #3: Yes

2. Has the statistical analysis been performed appropriately and rigorously?

Reviewer #1: Yes

Reviewer #2: Yes

Reviewer #3: Yes

3. Have the authors made all data underlying the findings in their manuscript fully available (please refer to the Data Availability Statement at the start of the manuscript PDF file)?

Reviewer #1: No

Reviewer #2: Yes

Reviewer #3: No

4. Is the manuscript presented in an intelligible fashion and written in standard English?

Reviewer #1: No

Reviewer #2: Yes

Reviewer #3: Yes

5. Review Comments to the Author

Reviewer #1: I am writing to request higher resolution figures for our paper to facilitate a smoother review process. The current resolution makes it difficult for reviewers to thoroughly evaluate the details.

I have already informed the editor about this issue, but I have not yet received the necessary documents. Your prompt assistance in providing these higher resolution figures would be greatly appreciated.

Reviewer #2: Manuscript Number: PGPH-D-24-01332

Article Type: Research Article

Full Title: Comparing lagged impacts of mobility changes and environmental factors on COVID-19 waves in rural and urban India: a Bayesian spatiotemporal modelling study

Short Title: COVID-19 drivers during two transmission waves in India.

Comments: This paper investigates the lagged impact of population movement changes and environmental factors on epidemic spread in urban and rural areas of India during two COVID-19 epidemic waves. The study used Bayesian spatio-temporal hierarchical modeling and Distributed Lag Nonlinear Modeling (DLNM) in conjunction with anonymized human mobility data provided by Google to analyze the drivers of transmission during the two epidemic waves in 2020 and 2021. The paper is well-organized, and the flow of ideas is generally coherent. Minor revisions particularly in the methodology, discussion of results, and policy implications, should improve the paper's clarity, coherence, and impact.

1. Introduction:

Expand the literature review: while the introduction gives a good overview, it could benefit from a broader discussion of existing research. Consider adding more recent studies (from 2022-2023) on the relationships between mobility, climate factors, and COVID-19, especially those using similar spatiotemporal modeling approaches in other regions.

Clarify the research gap: while the gap in previous studies is mentioned, more emphasis on why this study is critical in the global pandemic context, not just in India, would strengthen the rationale for the research.

2. Methodology:

The technical aspects of the Bayesian spatiotemporal hierarchical models and INLA are not well explained and should be stated clearly. How the drivers are included in the spatiotemporal hierarchical models and INLA. Briefly explain why certain variables (such as temperature and precipitation) were selected and how they contribute to the model.

Justify time lags: more explanation is needed about the rationale for choosing specific time lags (e.g., 0-3 weeks). Why were these particular lag intervals chosen? This can help reinforce the validity of the modeling approach.

3. Results:

Streamline the presentation: some parts of the results section are a little bit repetitive. Streamlining the discussion of the findings and emphasizing the most significant and novel results (especially regarding the role of mobility and intervention stringency) will improve readability.

Explain spatial random effects more clearly: the explanation of spatial random effects is a bit technical and could be clearer. A more intuitive description of what these effects represent (e.g., unmeasured regional differences) would help the reader understand their importance.

Reviewer #3: The authors examined the impact of mobility changes and environmental factors on COVID-19 transmission in India using the Bayesian spatiotemporal model. I have some comments.

1. I think the following sentence may be inappropriate: “However, no previous research has compared mobility patterns, or inter-district movement across both pandemic waves, relative to pre-pandemic mobility levels, and associated impact on COVID-19 transmission.”, since there have been studies comparing mobility patterns.

2. The authors mentioned that they used “pre-pandemic mean baseline levels of mobility for the first eight weeks of 2020”. Sensitivity analysis can be conducted to change “eight weeks” to others.

3. The authors mentioned that “Public holidays, which included the date of public holiday and one day before and after, were assigned a value of 1”. Sensitivity analysis can be conducted to change “the date of public holiday and one day before and after” to others.

4. I think the expression “the instantaneous basic reproduction number ( 0) over time” may be not appropriate, since 0 does not change over time.

5. How to determine the parameters of the gamma distribution for △ | _ (i.e., /0.5, 0.5)? The authors can provide references for this or give reasons.

6. The authors “explored exposure-lag response associations between the relative risk (RR) of increase in COVID-19 transmission, and changes in mobility…”. I think the expression “relative risk” is not appropriate, since △ is not a binary variable.

6. PLOS authors have the option to publish the peer review history of their article (what does this mean? ). If published, this will include your full peer review and any attached files.

**Do you want your identity to be public for this peer review?** For information about this choice, including consent withdrawal, please see our Privacy Policy .

Reviewer #1: No

Reviewer #2: No

Reviewer #3: No

---

## [Decision Letter · Decision Letter 1]

29 Jan 2025

PGPH-D-24-01332R1

Comparing lagged impacts of mobility changes and environmental factors on COVID-19 waves in rural and urban India: a Bayesian spatiotemporal modelling study

Dear Dr. Cleary,

Thank you for submitting your manuscript to PLOS Global Public Health. After careful consideration, we feel that it has merit but does not fully meet PLOS Global Public Health’s publication criteria as it currently stands. Therefore, we invite you to submit a revised version of the manuscript that addresses the points raised during the review process.

We look forward to receiving your revised manuscript.

Kind regards,

Sheikh Taslim Ali, M.Sc., Ph.D.

Academic Editor

Journal Requirements:

Additional Editor Comments (if provided):

Thank you for the revised manuscript. A quick minor but essential suggestion to revise the text before I am able to recommend for further process.

I suggest to revise the term 'the instantaneous basic reproduction number (R0) over time' to illustrate the basic reproduction number for each variant as measure of initial transmissibility (without the effects of interventions and no depletion in susceptibility in the population) as 'the variant-specific basic reproduction number (R0) across the waves'. The terms 'instantaneous' and 'over time' indicate the measure is temporal, where R0 has very specific meaning in epidemiology: R0 is a measure of intrinsic transmissibility.

In fact, during the emergence of these variants there were several interventions in force. In case, R0 counterfactually estimated assuming no intervention for emergence of each variant, we can't claim this measure is instantaneous. How have you estimated R0 when there were effects of interventions in the population already, based on the impact of intervention the population susceptibility changes? Otherwise, it can be termed as the variant-specific initial reproduction number (R_in) across the waves"? Where, R_in is a measure of initial transmissibility (not a temporal measure), still accounts the affects of intervention in the population.

Such coining of parameters over standard theory should be defined carefully. Therefore, please clarify or revise the text accordingly.

Reviewers' comments:

Reviewer's Responses to Questions

**Comments to the Author**

1. If the authors have adequately addressed your comments raised in a previous round of review and you feel that this manuscript is now acceptable for publication, you may indicate that here to bypass the “Comments to the Author” section, enter your conflict of interest statement in the “Confidential to Editor” section, and submit your "Accept" recommendation.

Reviewer #2: (No Response)

Reviewer #3: (No Response)

2. Does this manuscript meet PLOS Global Public Health’s publication criteria ? Is the manuscript technically sound, and do the data support the conclusions? The manuscript must describe methodologically and ethically rigorous research with conclusions that are appropriately drawn based on the data presented.

Reviewer #2: (No Response)

Reviewer #3: Yes

3. Has the statistical analysis been performed appropriately and rigorously?

Reviewer #2: (No Response)

Reviewer #3: Yes

4. Have the authors made all data underlying the findings in their manuscript fully available (please refer to the Data Availability Statement at the start of the manuscript PDF file)?

Reviewer #2: (No Response)

Reviewer #3: Yes

5. Is the manuscript presented in an intelligible fashion and written in standard English?

Reviewer #2: (No Response)

Reviewer #3: Yes

6. Review Comments to the Author

Reviewer #2: NA

Reviewer #3: The authors “explored exposure-lag response associations between the relative risk (RR) of increase in

COVID-19 transmission, and changes in mobility…”. I think the expression “relative risk” is not

appropriate. It can be changed to the ratio of XX.

7. PLOS authors have the option to publish the peer review history of their article (what does this mean? ). If published, this will include your full peer review and any attached files.

**Do you want your identity to be public for this peer review?** For information about this choice, including consent withdrawal, please see our Privacy Policy .

Reviewer #2: No

Reviewer #3: No

---

## [Editor Report · Decision Letter 2]

18 Feb 2025

Comparing lagged impacts of mobility changes and environmental factors on COVID-19 waves in rural and urban India: a Bayesian spatiotemporal modelling study

PGPH-D-24-01332R2

Dear Dr. Cleary,

We are pleased to inform you that your manuscript 'Comparing lagged impacts of mobility changes and environmental factors on COVID-19 waves in rural and urban India: a Bayesian spatiotemporal modelling study' has been provisionally accepted for publication in PLOS Global Public Health.

Best regards,

Sheikh Taslim Ali, M.Sc., Ph.D.

Academic Editor